Methods

# CRISPR/Cas9 bioluminescence-based assay for monitoring CFTR trafficking to the plasma membrane

Martin Ondra[1,4] , Lukas Lenart[1], Amanda Centorame[2,3], Daciana C Dumut[2,3], Alexander He[5], Syeda Sadaf Zehra Zaidi[5], John W Hanrahan[3,5] , Juan Bautista De Sanctis[1] , Danuta Radzioch[1,2,3] , Marian Hajduch[1,4,6]

**CFTR is a membrane protein that functions as an ion channel. Mutations that disrupt its biosynthesis, trafficking or function cause cystic fibrosis (CF). Here, we present a novel in vitro model system prepared using CRISPR/Cas9 genome editing with endogenously expressed WT-CFTR tagged with a HiBiT peptide. To enable the detection of CFTR in the plasma membrane of live cells, we inserted the HiBiT tag in the fourth extracellular loop of WT-CFTR. The 11-amino acid HiBiT tag binds with high affinity to a large inactive subunit (LgBiT), generating a reporter luciferase with bright luminescence. Nine homozygous clones with the HiBiT knock-in were identified from the 182 screened clones; two were genetically and functionally validated. In summary, this work describes the preparation and validation of a novel reporter cell line with the potential to be used as an ultimate building block for developing unique cellular CF models by CRISPR-mediated insertion of CF-causing mutations.**

## Introduction

Cystic fibrosis (CF) is a lethal, autosomal recessive disorder caused by mutations in the *CFTR* gene that results in aberrant transport of chloride, bicarbonate, and sodium ions (Kerem et al, 1989; Riordan et al, 1989; Rommens et al, 1989; Tabcharani et al, 1991; Poulsen et al, 1994; Stutts et al, 1995; Mall et al, 1996). CF affects the function of sweat glands, pancreas, liver, gastrointestinal tract, and reproductive and respiratory systems (Molina & Hunt, 2017). Despite the systemic nature of CF, recurrent lung infections are the major cause of morbidity and mortality (Turcios, 2020). Defective ions and fluid transport in the airway epithelia alter extracellular pH and lead to dehydration of the airway surface, translating into the impaired mucociliary clearance of thick, sticky mucus (Poulsen et al, 1994; Coakley et al, 2003; Boucher, 2007). These changes promote airway infections eliciting intense yet unresolved inflammation

(Pezzulo et al, 2012; Cantin et al, 2015). The vicious cycle of infection and inflammation leads to permanent structural airway damage and the development of bronchiectasis that, together with small airway obstruction, contributes to respiratory failure (Cantin et al, 2015; Turcios, 2020).

The *CFTR* gene spans ~190 kb on the long arm of chromosome 7 (7q31.2). Its 27 exons are translated into a 1,480-amino acid CFTR protein (Riordan et al, 1989; Tsui & Dorfman, 2013). CFTR, a member of the ATP-binding cassette protein superfamily, consists of two transmembrane domains, two nucleotide-binding domains (NBD), and a regulatory domain (RD). CFTR differs from the rest of the human ATP-binding cassette proteins in its structure and function, as it comprises a unique RD and operates as a cAMP-dependent ATP-gated ion channel (Sheppard & Welsh, 1999; Dean et al, 2001; Lopes-Pacheco, 2020). The transmembrane domains span through the plasma membrane (PM), each by six helical segments, and form the channel pore (Riordan et al, 1989; Sheppard & Welsh, 1999; Liu et al, 2017). The highly organized NBDs are localized in the cytosol with the unorganized RD containing recognition sequences for PKA and PKC phosphorylation (Rommens et al, 1989; Jia et al, 1997; Sheppard & Welsh, 1999). CFTR channel opening is induced by the dimerization of NBDs upon binding of two molecules of ATP. As the NBD1 is sterically blocked by the RD domain in the inactive state, both the regulatory insertion segment of NBD1, and the RD, need to be phosphorylated by PKA to enable the NBD dimerization. The ATP-dependent gating cycle is completed by the hydrolysis of ATPs returning the channel to its closed state (Sheppard & Welsh, 1999; Vergani et al, 2005; Sorum et al, 2017; Zhang et al, 2017; Della Sala et al, 2021).

With over 2,000 identified *CFTR* mutation variants (The clinical and Functional TRanslation of CFTR, 2023), the most common mutation is the deletion of three nucleotides leading to the loss of phenylalanine at position 508 of the CFTR protein (ΔF508). ~90% of people with CF (PwCF) carry at least one copy of ΔF508, with 50% being ΔF508 homozygotes (Guo et al, 2022). Conventionally, *CFTR* mutations are classified into six classes based on their effect on

[1]Institute of Molecular and Translational Medicine, Faculty of Medicine and Dentistry, Palacky University, Olomouc, Czech Republic   [2]Faculty of Medicine and Health Sciences, McGill University, Montreal, Canada   [3]RI-MUHC, Montreal, Canada   [4]Czech Advanced Technology and Research Institute, Palacky University, Olomouc, Czech Republic   [5]Physiology, McGill University, Montreal, Canada   [6]Laboratory of Experimental Medicine, Institute of Molecular and Translational Medicine, University Hospital Olomouc, Olomouc, Czech Republic

Correspondence: danuta.radzioch@mcgill.ca; marian.hajduch@upol.cz

CFTR biosynthesis and cellular phenotype (Zielenski, 2000; Boyle & De Boeck, 2013). Whereas the first two classes (I, II) are characterized by the absence of CFTR in the PM, the other four classes (III–VI) comprise *CFTR* mutations affecting channel function and/or the amount of CFTR protein rather than its trafficking. However, it has been described that some mutations result in a pleiotropic defect of the CFTR. For example, in the case of class II mutation ΔF508, most of the misfolded proteins do not reach the PM as they undergo endoplasmic reticulum-associated degradation (ERAD). Nonetheless, some of the misfolded ΔF508-CFTR can escape ERAD, thus reaching the PM, where they exhibit decreased activity and stability, both characteristic traits of class III and VI mutations (Veit et al, 2016).

As mentioned above, numerous mutations give rise to CF. Apart from conventional supportive therapy, focusing on alleviating CF symptoms, there has been a serious effort regarding the development of small molecules called modulators that can target and rectify the underlying CFTR defects. The most prominent categories of CFTR modulators are correctors and potentiators. Whereas correctors mediate the proper folding of mutated CFTR, thus enabling its glycosylation and membrane localization, potentiators increase the channel conductance and the open-state probability of CFTR already located in the PM (Rowe & Verkman, 2013; Lopes-Pacheco, 2016). Over the last two decades, a wide variety of HTS assays have been established to identify CFTR modulators. These assays can either address the channel function by directly or indirectly measuring CFTR ion flux (Ramalho et al, 2022) or assess CFTR localization via its epitope tagging (Carlile et al, 2007). The generation of a halide-sensitive YFP probe (Jayaraman et al, 2000; Galietta et al, 2001a) resulted in significant advancement in the field of functional HTS assays (Galietta et al, 2001b; Ma et al, 2002; Smith et al, 2017). Equally, the tagging of WT and ΔF508 CFTR extracellular loops with HA and FLAG tags (Howard et al, 1995; Carlile et al, 2007), without disrupting the protein folding and trafficking, opened a new chapter in the HTS identification of modulators that target defects in CFTR synthesis, processing, and subsequent membrane localization (class I and II mutations). Despite their unquestionable contribution to the development of CFTR modulators, the above-mentioned assays rely on the exogenous overexpression of mutated CFTR. This overexpression does not reflect the natural state of the cell and may lead to the disruption of the balanced stoichiometry of protein complexes and nonphysiological artifacts (Prelich, 2012; Moriya, 2015). As the exogenous mutated CFTR is expressed from cDNA, it does not capture the fundamental aspects of CFTR processing. Therefore, these models are inadequate for the identification of modulators targeting splicing mutations, intronic SNPs, and premature termination codons (Clancy et al, 2019).

Utilizing CRISPR/Cas9-mediated genome editing (Jinek et al, 2012; Cong et al, 2013) and the structural complementation reporter system HiBiT/LgBiT (Schwinn et al, 2018), we developed a bioluminescence-based assay for quantifying and localizing endogenously expressed WT-CFTR. The HiBiT/LgBiT is a bioluminescent binary reporter system derived from previously engineered NanoLuc luciferase (Hall et al, 2012). It takes advantage of the NanoLuc binary technology (NanoBiT; Dixon et al, 2016) exploiting the assembly of split NanoLuc from its 18-kD subunit called Large BiT (LgBiT) and a small complementary subunit. Several variants of small complementary subunits have been developed with various

affinities to LgBiT. One of them, an 11-amino acid peptide (1.3 kD), termed HiBiT (Schwinn et al, 2018), produces bright and quantitative luminescence upon high-affinity complementation with LgBiT in the presence of a substrate. Its small size makes HiBiT an ideal tag for monitoring endogenously expressed proteins when inserted by genome editing. The HiBiT-tagged proteins can easily be quantified and localized by "Add and Read" assays.

This study describes simple cell-based "Add and Read" assays for detecting endogenous levels of WT-CFTR and its membrane localization. CFTR expression, glycosylation, localization, and function are preserved after HiBiT knock-in into the fourth extracellular loop (ECL4) of WT-CFTR. siRNA-mediated knockdown of CFTR demonstrates the utility of our assays. Our validated cell-based model provides a platform for preparing new CF HTS models using the direct introduction of specific mutations by genome editing. Furthermore, our optimized pipeline for CRISPR/Cas9-mediated HiBiT knock-in enables the insertion of HiBiT into CFTR protein variants in other cell lines, allowing the creation of novel models for *CFTR* mutations with currently no available therapies.

# Results

## Assay design and principle

To develop novel bioluminescence-based assays for total quantification and live-cell detection of WT-CFTR in the PM, we used NanoBiT (Dixon et al, 2016) in combination with CRISPR/Cas9 gene editing. The NanoBiT is a unique fragment complementation reporter composed of two counterparts, HiBiT (1.3 kD) and Large Bit (LgBiT; 18 kD), derived from NanoLuc luciferase. We investigated two potential positions for CRISPR/Cas9-mediated knock-in of HiBiT in the ECL4 of WT-CFTR (Fig 1A) for bioluminescence-based quantification of both total endogenous levels (lytic assay) and live-cell detection of WT-CFTR in the PM (extracellular assay). In the lytic assay (Fig 1B), cells expressing WT-CFTR-HiBiT seeded in multi-well plates were lysed in a detergent-containing buffer in the presence of LgBiT and the substrate furimazine, followed by detection of luminescence. In contrast, no detergent was added in the reaction buffer when measuring live-cell plasma membrane localization in the extracellular assay (Fig 1B), so that only HiBiT-tagged proteins on the cell surface were detected. The luminescent signal in both assays was proportional to the amount of HiBiT-tagged CFTR in the cell lysate or in the PM of cells. Volumes were optimized for 96-well and 384-well plates for optimal assay performance, following a 1:1 ratio of medium (without FBS): Nano-Glo reagent for the lytic/extracellular assay, with volume ratios of 12.5:12.5 $\mu$l for 384-well and 50:50 $\mu$l for 96-well plates.

## Design and validation of Cas9/gRNA cleavage efficiency

Two potential insertion sites for the HiBiT tag were selected based on the limited availability of the protospacer adjacent motif (PAM) sequences in ECL4 (Fig 2A). These sequences are required for the Cas9 function. We designed four different CRISPR RNAs (crRNAs; Fig 2B), located directly upstream to the PAM sequences, targeting a

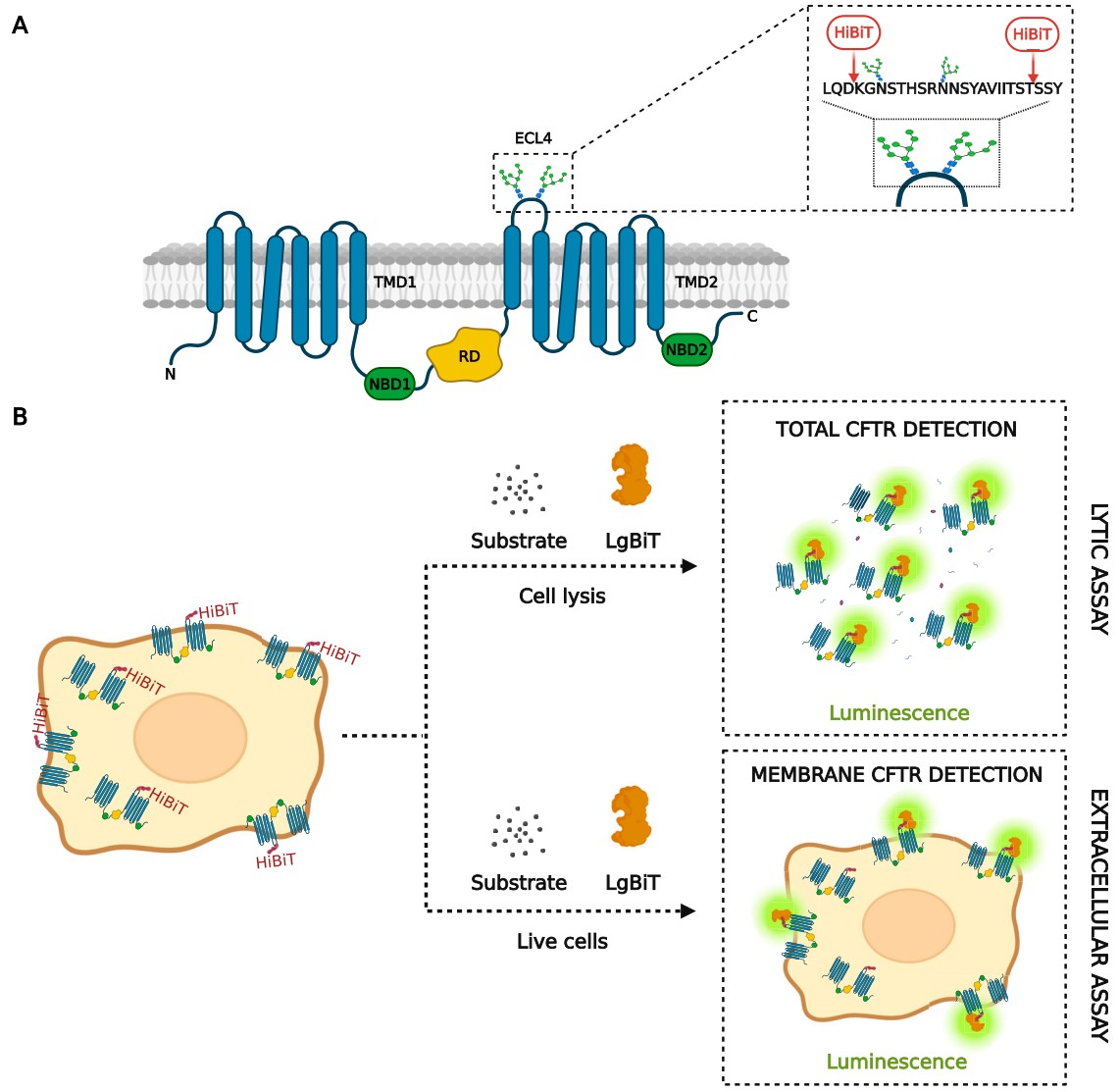

**Figure 1. Development and principle of HTS assays for WT-CFTR quantification and membrane detection.**
**(A)** Potential HiBiT tag positions in WT-CFTR. Schematic structure of CFTR domains: two transmembrane-spanning domains (TMD1 and 2), two nucleotide-binding domains (NBD1 and 2), and a regulatory domain (RD). Detail of the fourth extracellular loop (ECL4) depicting the CRISPR/Cas9 insertion of HiBiT tag at different positions, before or after glycosylation sites in the ECL4 of WT-CFTR. **(B)** Principle of Nano-Glo HiBiT detection systems. Quantification of total WT-CFTR-HiBiT expression (lytic assay) and live-cell detection in the membrane (extracellular assay).

region in *CFTR* locus chr7: 117,603,509–117,603,625 coding for ECL4. crRNA1 and crRNA2 targeted loci after the glycosylation (GLY) sites in ECL4, whereas crRNA3 and crRNA4 recognized regions before GLY sites. Together with a Cas9-specific trans-activating CRISPR RNA (tracrRNA), these crRNAs form guide RNA (gRNA). The transfection efficiency of electroporation with ribonucleoprotein (RNP) complexes (Cas9:gRNA) was verified by fluorescence microscopy 48 h after transfection (Fig S1A). 16HBE14o- cells (HBE) transfected with RNPs were positive for fluorescent signal because of the fluorophore ATTO 550 (ex 561 nm; em 620 ± 35 nm) attached to tracrRNA. Autofluorescence was negligible in mock electroporated cells (i.e., electroporated without RNP) or in non-electroporated cells. To precisely analyze gene-cleavage efficiency (double-stranded breaks) and repairs by nonhomologous end joining (NHEJ) in the pool of

edited cells, the targeted region in the genomic DNA was analyzed by Sanger sequencing 72 h posttransfection. Using Tracking of Indels by Decomposition (TIDE) online web tool software (Brinkman et al, 2014), we identified crRNA1 and crRNA3 as the best candidates, with 53.6% and 56.4% total gene-cutting efficiencies (Fig S1B). They were used in subsequent experiments for template-dependent HiBiT insertion using the homology-directed repair (HDR) mechanism of cells. For both crRNAs, the indels caused by NHEJ were found in proximity to the predicted Cas9 cut site of the PAM sequence.

### HDR-mediated knock-in of HiBiT tag into ECL4

To insert HiBiT into ECL4, we performed gene editing of the HBE cells using two previously validated RNP complexes in the presence of

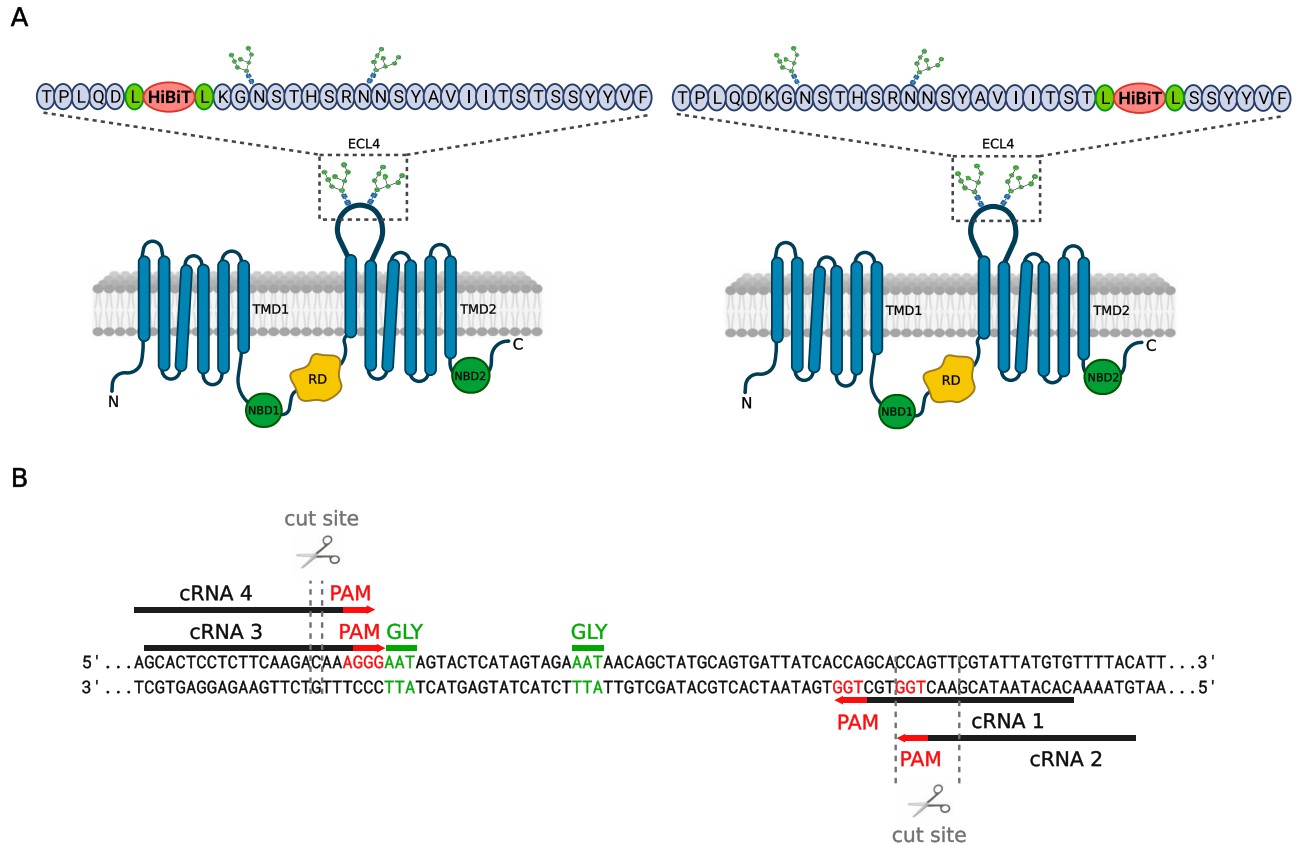

**Figure 2. HiBiT tag positions in ECL4 of WT-CFTR and schematic representation of crRNAs sequences.**
**(A)** Two variants of HiBiT tag knock-in positions in ECL4. **(B)** crRNA target sequences with protospacer adjacent motif (PAM) sequence in red, glycosylation sites (GLY) in green, and cleavage site of each crRNA.

synthetic single-stranded oligodeoxynucleotide (ssODN) donors (Fig 3A). Either RNP1 complex containing crRNA1 or RNP3 complex, with crRNA3, targeting loci before or after the GLY sites in ECL4, was used. We tested ssODNs, with 6 or 8 amino acid linkers (6 or 8AA) on either side of the HiBiT sequence, specific for each RNP complex (Fig 3). ssODNs were designed with asymmetric lengths of homology arms to increase HDR efficiency and silent mutations in the PAM sequence to prevent recutting of the modified locus (Table S1). To assess the effect of the AA linker's length on HiBiT detection and the stability of the luminescent signal, lytic (Fig 3B) and extracellular (Fig 3C) assays were performed 7 and 14 d after cell electroporation (Figs 3B and C and S2A and B). The luminescent signal was recorded for 3 h to show the kinetic properties of both assays. The lytic and extracellular assays were performed with various amounts of seeded cells to determine an optimal cell number for the assays. Pools of cells edited with RNPs and ssODN with 8AA linkers provided enhanced signal compared with RNPs with 6AA linkers, when used for both lytic (RNP1/8AA $8.5 \times 10^3$ RLU, RNP3/8AA $12.5 \times 10^3$ RLU; Fig 3B) and extracellular assays (RNP1/8AA $5.5 \times 10^3$ RLU, RNP3/8AA $6.3 \times 10^3$ RLU; Fig 3C). Unmodified HBE cells (control) had a very low background auto-luminescence ~$1 \times 10^3$ RLU (relative light unit) in the lytic assay and ~$1.3 \times 10^3$ RLU in the extracellular assay (highest concentration of cells seeded). Consequently, we proceeded with RNPs and ssODNs containing 8AA linkers for further experiments.

## Preparation of monoclonal cell lines expressing WT-CFTR-HiBiT

Limiting dilution cloning was used (Fig 4A) to prepare a monoclonal cell line from pools of cells edited either with RNP1/8AA or RNP3/8AA ssODN. Out of 182 tested clones in the primary screening (91 clones, A-clones RNP1/8AA; 91 clones, B-clones RNP3/8AA), we identified 32 positive clones (10 A-clones and 22 B-clones) with varying levels of luminescence (Fig 4B). In the secondary screening, we retested 27 out of the 32 clones (Fig 4C) to confirm the hits from the primary screening and to identify those expressing WT-CFTR-HiBiT in the PM. All clones found in the primary screening using lytic assay were positive in the secondary screening. At the same time, in the extracellular assay, only eight clones (A22, A32, A37, A62, A67, B6, B38, B60) had notably superior signals to that of non-modified HBE cells (Fig 4C). Moreover, we used PCR genotyping (Fig 4D) to distinguish heterozygous from homozygous clones. Heterozygotes were represented by two bands (849 and 930 bp) corresponding to unedited and edited alleles (+81 bp) of *WT-CFTR*. In contrast, only one band (930 bp) was detected for homozygous clones. Overall, we identified nine homozygotes (A22, A65, A67, A83, B38, B42, B57, B60, B89). The clones that also generated luminescence in the extracellular assay were sequenced for validation of the correct HiBiT knock-in position. We identified three clones (A22, A67, and B38) with a correct knock-in of the HiBiT sequence containing 8AA

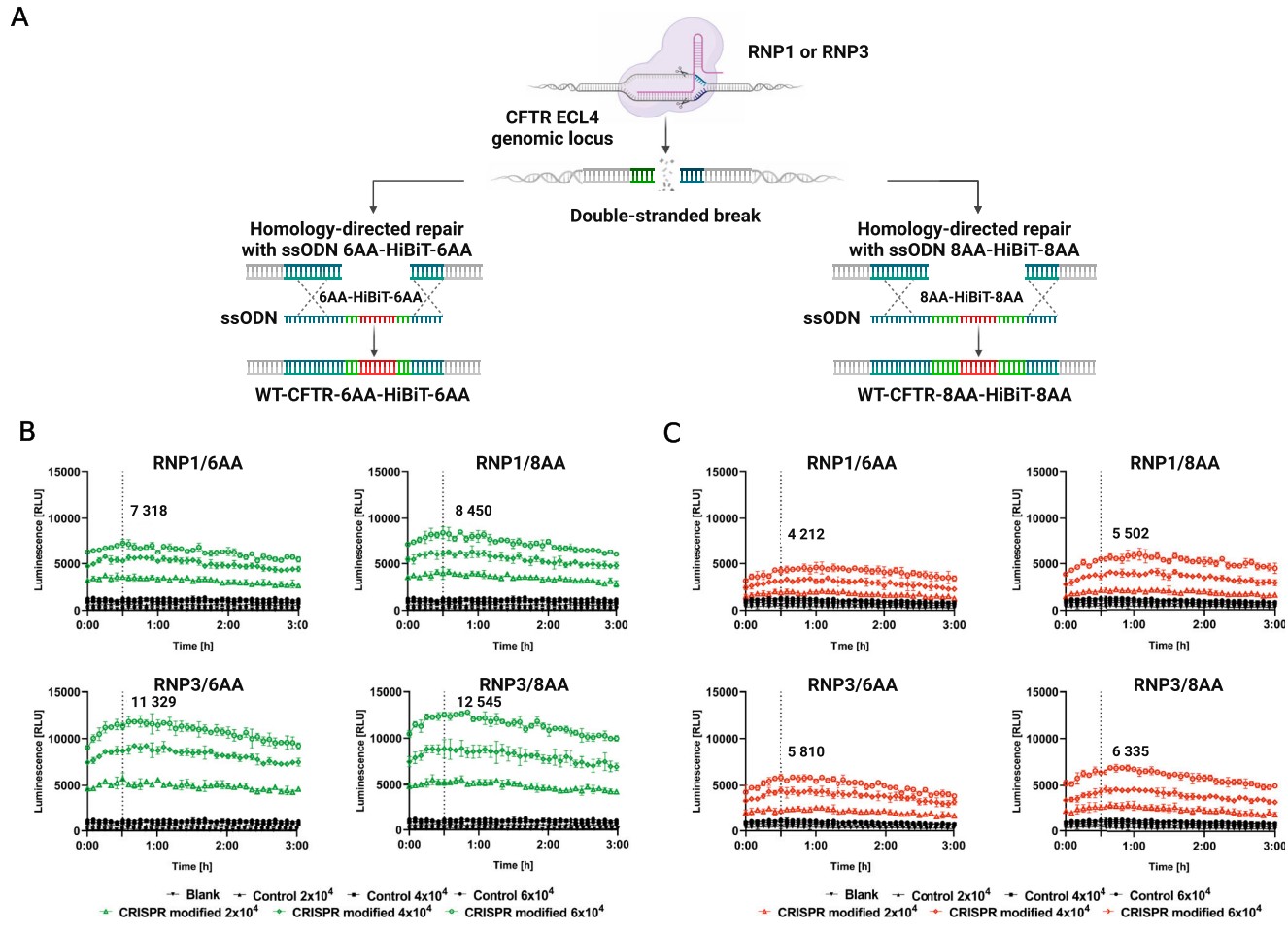

**Figure 3. Comparison of single-stranded oligodeoxynucleotide with 6 and 8AA linkers for HiBiT knock-in.**
**(A)** Schematic of homology-directed repair-mediated knock-in of HiBiT with either 6 or 8AA linkers. **(B)** Lytic assay and **(C)** Extracellular assay 7 d posttransfection. Three different concentrations of cells (2 × 10³, 4 × 10³, and 6 × 10³) were seeded. Highlighted numbers in graphs represent the signal for the highest concentration of cells after 30 min (mean ± SD, n = 3, technical replicates). Control = unmodified HBE cells.

linkers; an example is illustrated in Fig 4E (clone B38). As the clone A67 lost its phenotypic properties during further cultivation, only clones A22 and B38 were analyzed for potential CRISPR/Cas9 gene editing off-target effects (Table S2) using Sanger sequencing. There were no variations detected in either of the two clones compared with non-modified HBE cells (Fig S3A and B).

## Validation of WT-CFTR-HiBiT glycosylation, localization, and function

To exclude possible alteration of protein processing and glycosylation status of WT-CFTR-HiBiT caused by HiBiT knock-in, Western blotting (WB) and HiBiT blotting (HB) analysis were done. When compared with HBE parental cells, A22 and B38 clones demonstrated similar patterns for CFTR band C (~170 kD, complex-glycosylated CFTR) and band B (~150 kD, core-glycosylated CFTR present in the ER) in WB analysis (Fig 5A). However, a lower expression of CFTR in A22 and B38 clones was detected. Whereas the HB analysis revealed an additional band in the vicinity of band B in clone A22, clone B38 did not exhibit any additional bands in HB or WB analysis (Fig 5A), confirming the

insertion and specific expression of HiBiT exclusively in WT-CFTR, and unaltered glycosylation of the WT-CFTR-HiBiT. Next, the localization of WT-CFTR-HiBiT was assessed by immunofluorescence (IF) detection using confocal microscopy. The expression and localization of WT-CFTR-HiBiT in the B38 clone were compared with WT-CFTR in parental HBE cells by immunostaining with or without permeabilization of cells (Fig 5B). The monoclonal antibody used in IF binds to the first extracellular loop (ECL1) of CFTR, enabling detection of CFTR in the PM and throughout the cytoplasm with cell permeabilization. As a result, we observed invariable membrane staining of WT-CFTR in HBE compared with WT-CFTR-HiBiT in B38 cells, respectively. A notably higher level of immunofluorescence signal was observed for conditions with permeabilization corresponding to the sum of stained CFTR in the PM and the cytoplasm. To validate the correct anion channel activity of WT-CFTR-HiBiT protein, Ussing chamber measurements were performed. First, amiloride was added to ensure that sodium absorption did not contribute to the measured short-circuit current ($I_{SC}$). Sequential administration of forskolin and genistein caused maximal activation of CFTR channels in the apical membrane of the cell monolayer, thus elevating $I_{SC}$. The CFTR

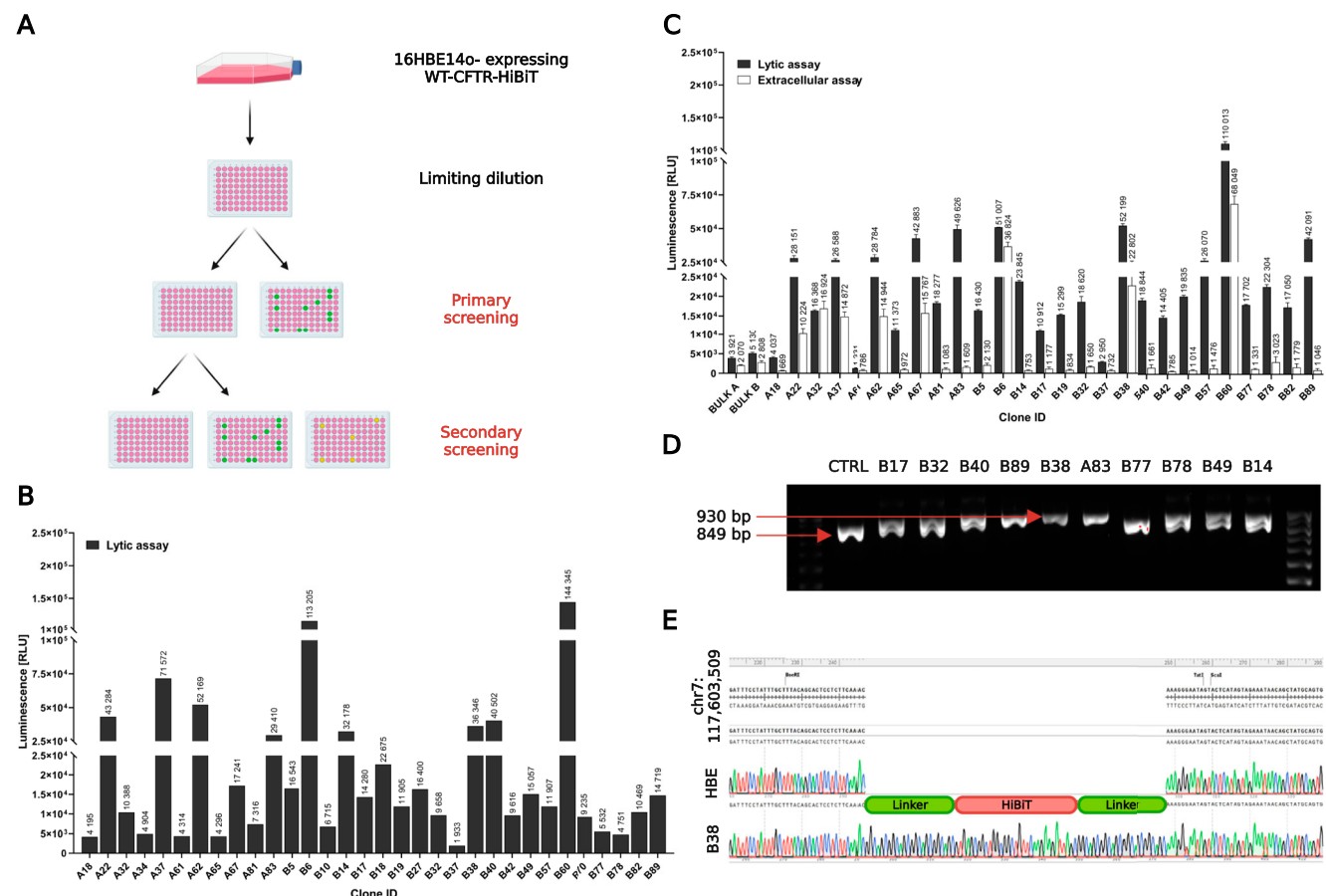

**Figure 4. Generation of monoclonal cell lines expressing WT-CFTR-HiBiT and validation of HiBiT tag knock-in position.**
**(A)** WT-CFTR-HiBiT–positive clones identification workflow. The pools of cells edited either with RNP1/8AA or RNP3/8AA single-stranded oligodeoxynucleotide (template for HiBiT tag knock-in) were used for limiting dilution cloning to obtain monoclonal cell lines that were subjected to primary and secondary screenings. **(B)** Primary screening. Lytic assay to detect positive clones for HiBiT. **(C)** Secondary screening. Positive clones from primary screening were retested using the lytic and extracellular assays to detect total and membrane WT-CFTR-HiBiT expression (mean ± SD, n = 3, technical replicates). **(D)** PCR genotyping. The representative electrophoresis separation pattern of PCR products. Heterozygotes are represented by two bands (849 and 930 bp) and homozygotes by only one band (930 bp). CTRL = non-modified HBE cells (849 bp). **(E)** Sanger sequencing. Chromatogram of the B38 clone compared with parental HBE cells and reference *CFTR* genomic sequence.

inhibitor CFTR$_{Inh}$-172 (Inh172), decreased I$_{SC}$ to its baseline revealing the total contribution of CFTR channel to the measured I$_{SC}$ (ΔI$_{SC}$). This decrease was used to quantify CFTR activity (Fig 5C). Although ΔI$_{SC}$ recorded for clone B38 was lower than for HBE, B38 still exhibited an adequate response to forskolin and to inhibitor Inh172; representative short-circuit current traces are illustrated in Fig 5C. We further confirmed that all measured monolayers were "non-leaky" by measuring transepithelial electrical resistance (TEER; Fig 5C). Only monolayers with a TEER ≥ 200 Ω × cm$^2$ were included in the analysis. These results demonstrated that the knock-in of HiBiT into the ECL4 did not impair the glycosylation, localization or function of WT-CFTR.

## Down-regulation of CFTR expression is detectable by lytic and extracellular assay

To evaluate the ability of our assays to quantify total and membrane CFTR, we treated clone B38 with three unique siRNAs, effectively reducing CFTR levels after 48 and 72 h of treatment according to both lytic and extracellular assays. We observed a maximal reduction of

CFTR expression in cells transfected with siRNA A, after 72 h. Namely, in cells treated with 50 nM siRNA A, a 58% decrease in lytic (P < 0.001; Fig 6A) and a 52% decrease in the extracellular assays (P < 0.001; Fig 6A) were detected when compared with non-treated cells. No significant difference was observed between non-treated cells and those transfected with scrambled siRNA. To exclude potential cytotoxicity effects of siRNA, that might falsely affect the level of luminescence, a cell viability (MTS) assay was performed. Clone B38 was treated with siRNA in parallel to the lytic and extracellular assays, in the same manner. MTS assay revealed no significant effect of siRNA treatment on cell viability (Fig 6B). In summary, siRNA-mediated silencing results reflect the capacity of our assays to detect and quantify changes in total WT-CFTR-HiBiT after lysis, and in expression of WT-CFTR-HiBiT on the surface of live cells.

## Monitoring of CFTR trafficking to the plasma membrane

To further validate the applicability of our assays in monitoring CFTR trafficking to the plasma membrane, we subjected B38

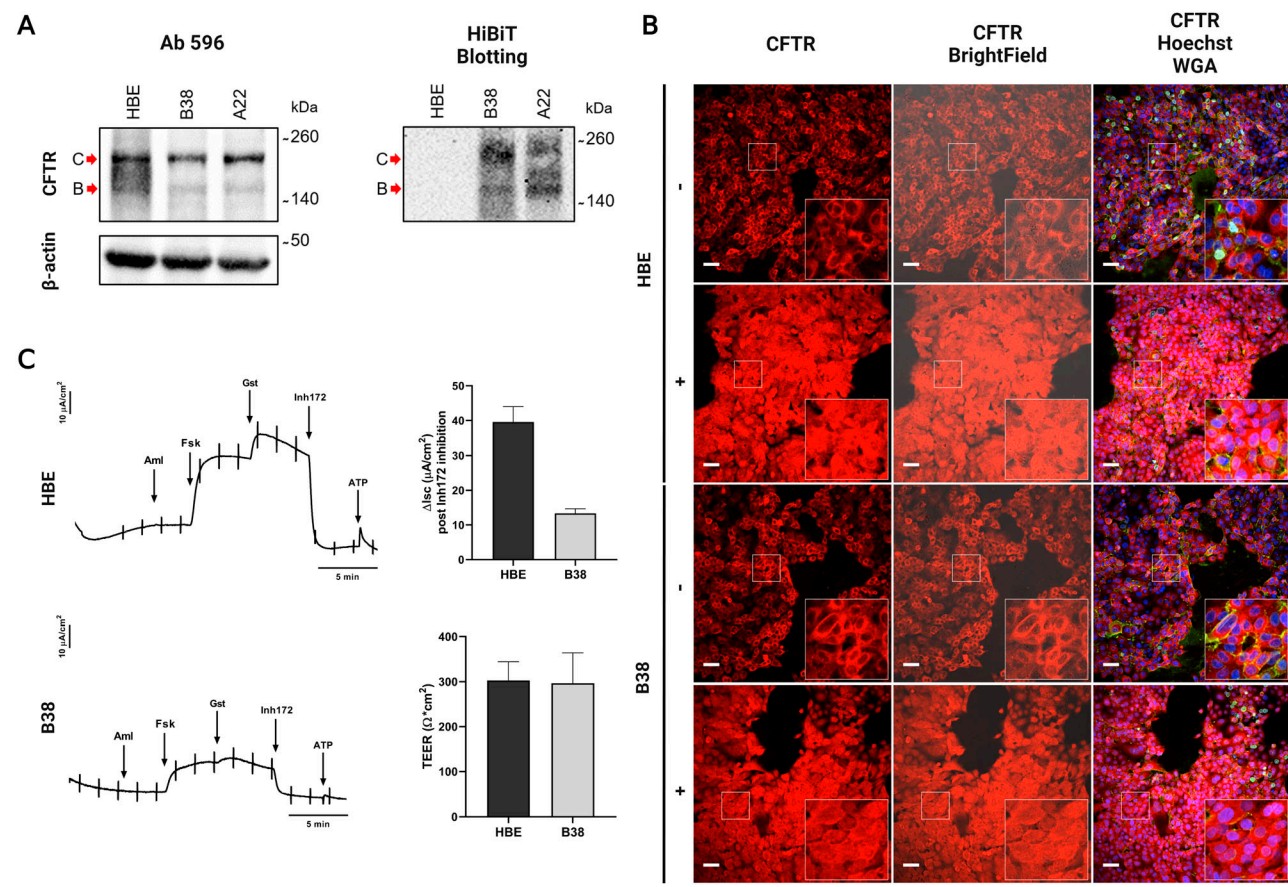

**Figure 5. Validation of clones.**
**(A)** Western and HiBiT blotting. WB immunoblot probed with antibody 596 against the NBD2 domain of CFTR. C-band indicates the fully-glycosylated form of WT-CFTR ~170 kD, and the B-band, the ER core-glycosylated form ~150 kD. **(B)** Immunocytochemistry. Detection of CFTR by confocal microscopy in HBE and B38 cells (20x objective). Cells incubated with TJA9 anti-CFTR antibody against ECL1 of CFTR and thereafter with a secondary antibody conjugated to Alexa 647 (red). Nuclei stained with Hoechst 33342 (blue) and PM with wheat germ agglutinin (WGA; green). Zoomed-in insets at 3× magnification (zoomed-in factor). Scale bar: 50 μm. **(C)** Short-circuit current—Ussing chambers. Short-circuit current was recorded after exposure to amiloride (Aml; 10 μM), forskolin (Fsk; 10 μM), genistein (Gst; 50 μM), ATP (10 μM), and current inhibitor CFTR$_{Inh}$-172 (Inh172; 10 μM). Maximal short circuit current ($\Delta I_{SC}$) was calculated post-forskolin, genistein activation, and current inhibition (Inh172). Only monolayers with transepithelial electrical resistance ≥ 200 Ω × cm² were included in the analysis. Graphs show mean $I_{SC}$ with SD. All experiments were done in three independent biological replicates (n = 3).
Source data are available for this figure.

cells to RAB5 and RAB11 siRNA treatments to affect the endocytic trafficking of CFTR (Fig 7A). After RAB5 siRNA treatment, we observed a substantial increase in the luminescent signal within cells. In contrast, treatment with RAB11 siRNA led to a signal decrease (Fig 7B). Notably, there was no significant impact of siRNA treatment on cell viability. In addition, besides the specific down-regulation of expression of aforementioned RABs, we investigated the impact of Brefeldin A, a compound affecting the intracellular transport of proteins from the endoplasmic reticulum (ER) to the Golgi apparatus (GA; Fig 7A). Our observations revealed a dose-dependent reduction in CFTR levels after a 6-h Brefeldin A treatment, with no notable toxicity up to 12.5 μM (Fig 7C). In essence, the results presented here firmly affirm the potential of our assays not only in monitoring the overall CFTR protein levels but also in evaluating its trafficking and membrane localization.

## Discussion

Various HTS assays have been developed for monitoring CFTR protein expression and trafficking and have been leveraged to discover new CFTR modifiers. The overexpression of CFTR tagged either with intracellular or extracellular epitopes such as FLAG-tag, HA-tag (Howard et al, 1995) or fluorescent tags (Moyer et al, 1998) was employed to study CFTR trafficking, its subcellular localization, and degradation, using fluorescent microscopy (Meacham et al, 2001; Carlile et al, 2007). Although sophisticated, these approaches exploit the overexpression of recombinant proteins which can produce an imbalance in protein processing, localization, and levels, leading to nonphysiological artifacts (Prelich, 2012; Moriya, 2015). In the present study, we described the development and validation of novel assays for detecting total and plasma membrane WT-CFTR protein. Our goal was to develop a system with

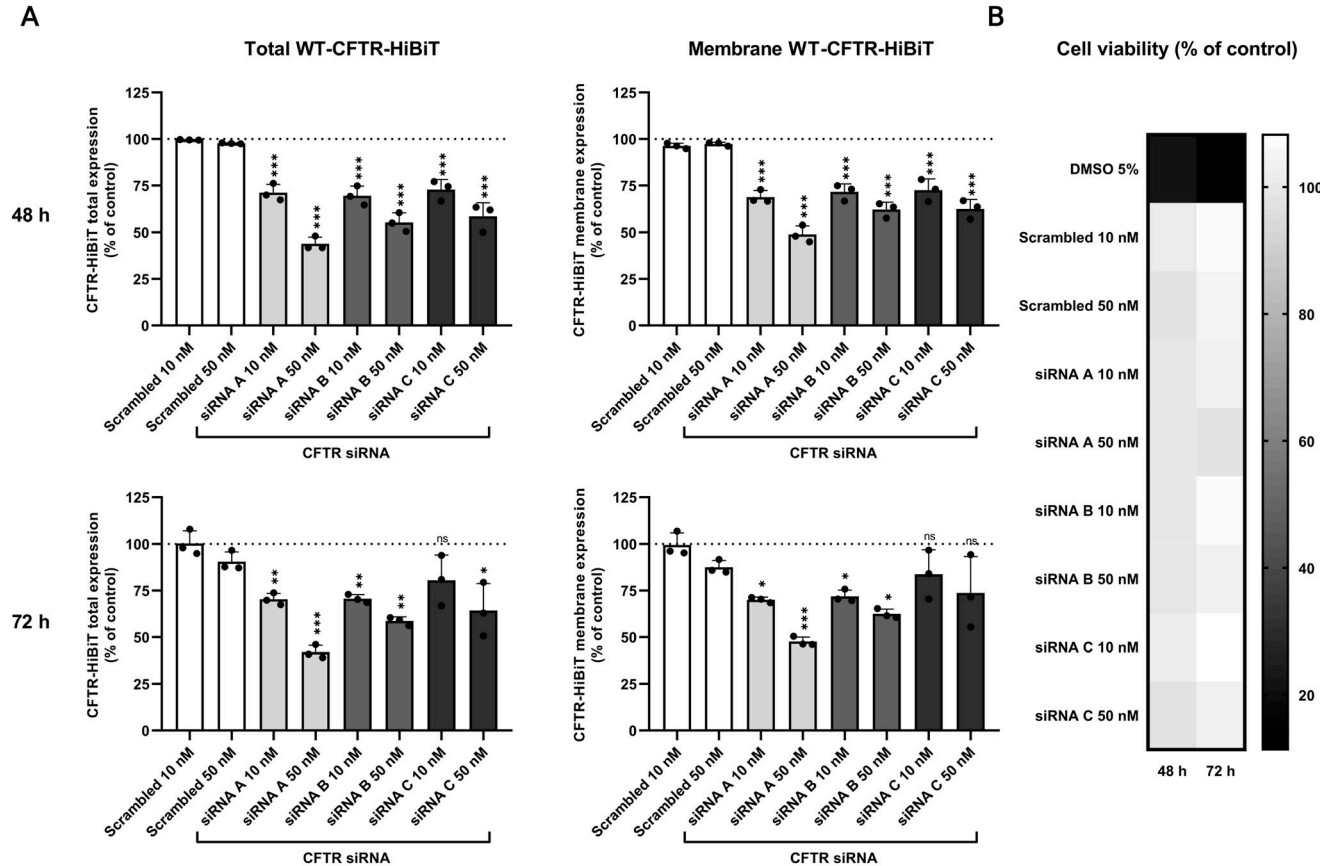

**Figure 6. Down-regulation of CFTR expression in the B38 clone using siRNA.**
**(A)** Lytic and extracellular assays after 48 and 72 h of siRNA treatment. The luminescent signal was measured after 30 min in both assays (mean ± SD). **(B)** Cell viability. The effect of siRNA on cell viability was measured by MTS assay (mean). All experiments were done in three independent biological replicates (n = 9; technical replicates). *P*-values were calculated using one-way ANOVA-Tukey's test, ns = nonsignificant, *P ≤ 0.05, **P < 0.005, ***P < 0.001.

endogenous expression of WT-CFTR from its natural genome locus that could be applicable for further development of CF model systems. We used CRISPR/Cas9-targeted gene editing with the HiBiT tag (Schwinn et al, 2018) to establish new bioluminescence-based reporter assays (Fig 1B).

Since the recognition of CRISPR/Cas9's potential to allow programmable genome editing (Jinek et al, 2012; Cong et al, 2013), disease-specific model system engineering has been expanding (Dow, 2015; Kampmann, 2020). CRISPR-derived models help elucidate disease mechanisms and identify potential therapeutic targets. To target specific DNA sequences for Cas9 endonuclease cleavage, a gRNA is required. crRNA, a part of gRNA that is complementary to the desired DNA sequence, directs the entire RNP complex to generate site-specific double-strand DNA breaks (DSB; Jinek et al, 2012; Cong et al, 2013; Mali et al, 2013; Konstantakos et al, 2022). Therefore, we designed and tested four crRNAs targeting ECL4 in WT-CFTR (Fig 2B) to find the most efficient one. Through electroporation-mediated delivery of RNP complexes into the 16HBE14o- cells and subsequent analysis of gene-editing events by TIDE online web tool software (Fig S1B), we identified crRNA1 and crRNA3 as the best candidates for further genome editing.

DSB created by the CRISPR/Cas9 complex can be repaired through NHEJ or HDR mechanisms (Kim & Kim, 2014). In the absence of a homologous DNA template, DSB is repaired by NHEJ creating small insertions or deletions at targeted loci. On the other hand, in the presence of a homologous template, DSB can be correctly repaired by HDR. The natural HDR mechanism of cells can be applied to introduce any desired base-pair changes or insertions using synthetic homologous repair templates (Cong et al, 2013; Mali et al, 2013). In our study, HDR-mediated knock-in of the HiBiT tag was performed. Based on the principle of NanoBiT technology (Dixon et al, 2016), HiBiT and LgBiT form an active luciferase enzyme, allowing for the quantification of endogenous, low-abundance proteins by luminescence detection (Schwinn et al, 2018). The spatial accessibility of HiBiT for LgBiT binding is a crucial factor. To optimize the LgBiT to HiBiT binding, we added 6 or 8AA linkers around the HiBiT sequence in the donor ssODN template to protrude HiBiT from the ECL4 rigid structure (Fig 3A). In the pools of edited cells, 8AA linkers provided an enhanced luminescent signal compared with cells edited by RNPs in the presence of ssODNs with 6AA linkers (Fig 3B and C).

A consistent genetic background of isogenic (monoclonal) cell lines is one of the essential factors for preparing relevant disease models, especially for CF (Pedemonte et al, 2010). Phenotype–genotype studies in PwCF showed that the same CF-causing mutations could have a distinct disease outcome based on the genetic makeup of each person (Zielenski, 2000; Cutting, 2010, 2015). Thus, to develop reproducible cell-

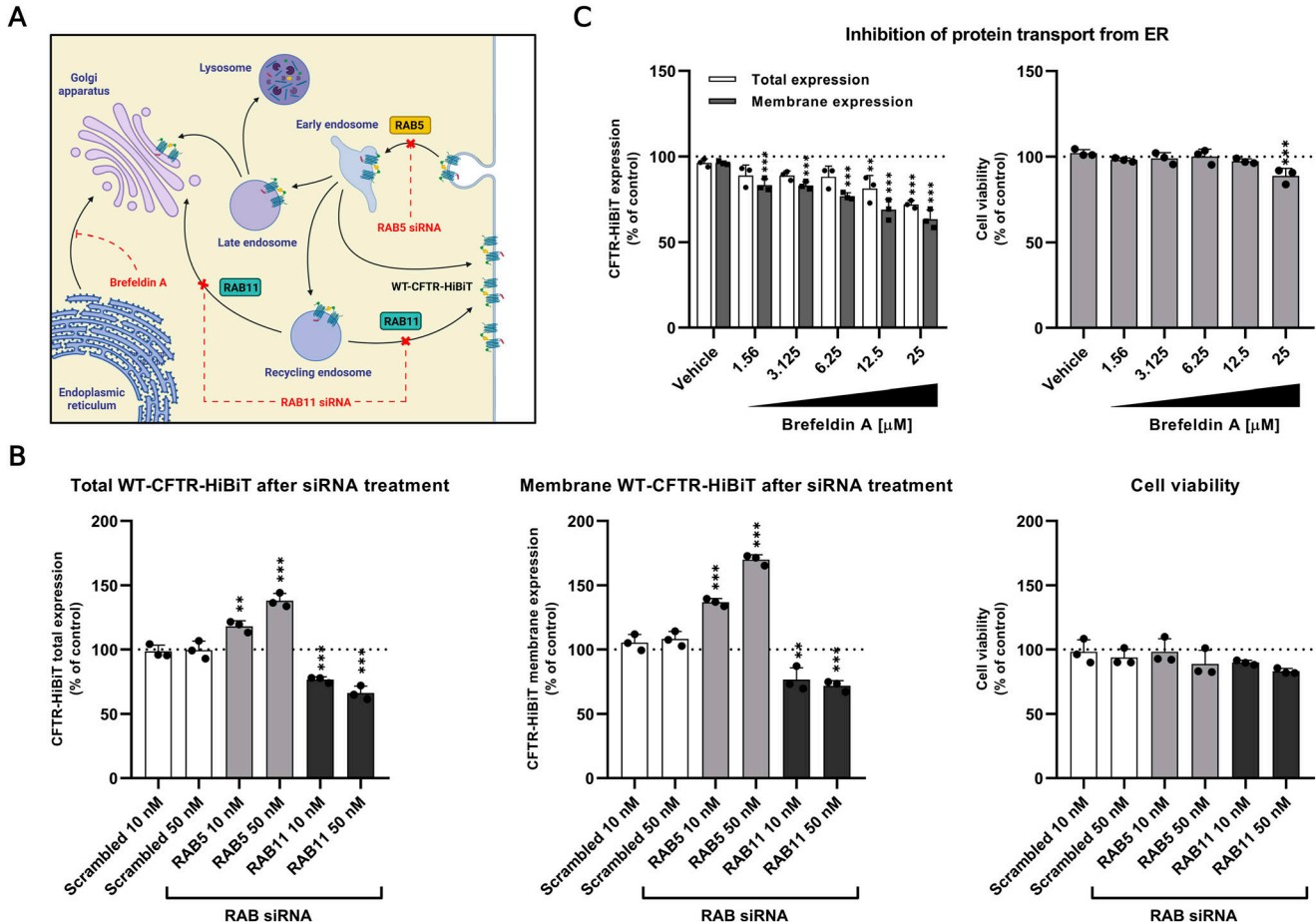

**Figure 7.  Modulation of CFTR trafficking.**
**(A)** Diagram of RAB5, RAB11 siRNA, and Brefeldin A effect on CFTR trafficking. **(B)** RAB5 and RAB11 siRNA treatments. Lytic and extracellular assays were performed after 48 h of siRNA treatments. **(C)** Brefeldin A inhibition of protein transport from ER to GA. Lytic and extracellular assays were performed after 6 h treatment. The effect of siRNA and Brefeldin A treatments on viability was measured by MTS assay (mean ± SD). The luminescent signal was measured after 30 min (mean ± SD). All experiments were done in three independent biological replicates (n = 9; technical replicates). *P*-values were calculated using one-way ANOVA-Tukey's test, ns = nonsignificant, *$P ≤ 0.05$, **$P < 0.005$, ***$P < 0.001$.

based systems for drug discovery screening, a relevant material for the introduction of disease-causing mutations is crucial. To address this, we have carried out limiting dilution cloning to obtain HiBiT-positive monoclonal cell lines. Namely, in two rounds of clone screening (Fig 4A–C), we identified 27 clones positive for HiBiT knock-in. In addition to that, ~5% of all clones tested (9 of 182) were identified as homozygotes for HiBiT knock-in using PCR genotyping (Fig 4D), which is in accordance with previously reported data for HDR efficiency in the presence of synthetic homologous repair templates (Ruan et al, 2019).

Once monoclonality has been assessed, we further characterized two candidate isogenic cell lines (B38 and A22). CFTR has two N-linked glycosylations at positions 894 and 900 (Riordan et al, 1989; Cheng et al, 1990; Gregory et al, 1990). As previously reported, disruption of these glycosylation consensus sequences reduces WT-CFTR stability (Chang et al, 2008). With this in mind, the effect of HiBiT tag insertion into two distinct sites in ECL4, either before (B38 clone) or after GLY sites (A22 clone), was evaluated. As confirmed by our results, we detected two WT-CFTR-HiBiT bands corresponding to core glycosylated (band B) and fully glycosylated (band C) CFTR in

conventional WB (Fig 5A). In WB, the C band was the most prominent in both clones, confirming the non-aberrant CFTR glycosylation process after the HiBiT knock-in. On the contrary, in HB, an additional band was detected for clone A22. Thus the B38 clone was selected for final validation. In addition, we also confirmed the accurate localization and trafficking of WT-CFTR-HiBiT in the PM by fluorescence microscopy (Fig 5B). Importantly, we did not find significant differences in immuno-staining patterns between parental HBE cells and the B38 clone. The last step of functional validation was carried out by short-circuit current measurements. HBE are immortalized airway-derived epithelial cells, which when confluent, become polarized and form tight junctions, thus manifesting strong transepithelial electrical resistance and maintaining cAMP- and Ca²⁺-dependent Cl⁻ currents (Cozens et al, 1994). The clone B38 showed a lower $\Delta I_{SC}$ compared with the HBE parental cells but still was adequately stimulated by forskolin and inhibited by CFTR$_{Inh}$-172. The reduced $\Delta I_{SC}$ of the B38 clone was presumably caused by lower expression of total CFTR (Fig 5A), most probably generated by clonal selection. We observed a similar TEER (Fig 5C)

for HBE and clone B38 indicating normal barrier function. These observations are consistent with previous reports (Cozens et al, 1994; Illek et al, 2008; Gianotti et al, 2018). To conclude, the HiBiT insertion before GLY sites in ECL4 did not disrupt WT-CFTR glycosylation, localization or function.

To validate the simple "Add and Read" format of the lytic and extracellular assays, the total and the plasma membrane WT-CFTR-HiBiT were quantified after knockdown by siRNAs. CFTR siRNA specifically decreases the amount of CFTR mRNA available for translation (Elbashir et al, 2001), thus reducing the level of total and membrane CFTR. In summary, treatment of the B38 clone with three variants of siRNAs significantly decreased both total WT-CFTR-HiBiT up to 58% and plasma membrane CFTR up to 52% (Fig 6A). In addition, we did not detect any cytotoxicity of the siRNA (Fig 6B). Our observations support the utility of our assays for the direct detection of changes in WT-CFTR levels and PM localization.

The level of membrane CFTR results from balanced CFTR expression, degradation, membrane delivery, and its endocytic recycling. One of the essential factors in the endocytic recycling of CFTR are the small RAB GTPases that function as molecular switches regulating the effector proteins that mediate intracellular trafficking (Farinha & Canato, 2017). To assess the correct trafficking and subsequent membrane localization of WT-CFTR-HiBiT in the B38 cell line, we specifically down-regulated the expression of RAB5 and RAB11 by RNA interference. Because RAB5 promotes the internalization of CFTR from PM into early endosomes and RAB11 regulates the trafficking of CFTR from the recycling endosomes to PM and GA (Ameen et al, 2007), the down-regulation of the RAB5 led to increased membrane CFTR levels, whereas the down-regulation of the RAB11 resulted in decreased membrane CFTR (Fig 7B). Moreover, the suitability of our novel system for monitoring of CFTR membrane trafficking was corroborated by disrupting CFTR trafficking from ER to GA by Brefeldin A as described by Donaldson et al (1992).

As previously reported (Zhang et al, 2023), HiBiT tagging technology is ideal for high-throughput applications. The lytic and extracellular assays were readily transferred from a 96-well plate to the 384-well plate reader format, showing little variation between replicates (Fig S4A–D).

Altogether, we have developed cell-based bioluminescence assays that are convenient and scalable for high-throughput determination of the total or live-cell membrane localized WT-CFTR. Our validated model can potentially be used as a platform for the preparation of new high-content and/or high-throughput screening CF models for specific *CFTR* mutations, including rare mutations that are difficult to study in primary cells. Moreover, our CRISPR-based pipeline for CFTR HiBiT tagging can be used in other cell lines expressing specific mutations in the *CFTR* gene.

# Materials and Methods

## Reagents

Alt-R CRISPR-Cas9 reagents (S.p. HiFi Cas9 Nuclease V3, Electroporation enhancer, tracrRNA ATTO550, and crRNAs) and ssODN (HDR Donor Oligos) were purchased from Integrated DNA Technologies. PCR primers were obtained from GeneriBiotech. Nano-Glo HiBiT Lytic Detection System and Nano-Glo HiBiT Extracellular Detection System were purchased from Promega. Antibodies: mouse anti-human CFTR antibodies (569, TJA9; CF Foundation), donkey anti-mouse IgG (H + L) Highly Cross-Adsorbed Secondary Antibody (Alexa Fluor 647; Invitrogen). siRNA: CFTR Human siRNA Oligo Duplex (Locus ID 1080; Origene), RAB5 (RAB5A) Human siRNA Oligo Duplex (Locus ID 5868), and RAB11 Human siRNA Oligo Duplex (Locus ID 8766).

## Cell culture

Human bronchial epithelial cells (16HBE14o-) were kindly provided by Dr. Dieter Gruenert (UCSF). 16HBE14o- cell line was originally derived from the surface epithelium of bronchi from a 1-yr-old male heart–lung transplant patient and immortalized by calcium phosphate transfection with the pSVori- plasmid (Cozens et al, 1994). Cells were cultured in complete Eagle's minimal essential medium (Enzo) with 10% FBS, 100 U/100 µg per ml penicillin/streptomycin (Gibco) under 5% $CO_2$ at 37°C.

## CRISPR gRNA preparation, RNP complex formation, and electroporation

All crRNAs were designed using CHOPCHOP software (Labun et al, 2019). crRNA sequences can be found in Table S1. crRNA (200 µM) and tracrRNA (200 µM) were mixed in a 1:1 ratio and incubated at 95°C for 5 min to prepare gRNA. RNP complexes for one transfection ($2.4 \times 10^5$ cells) were formed by mixing 3 µg of Alt-R S.p. HiFi Cas9 Nuclease V3 protein and 0.5 µl of 100 µM gRNA for 15 min at RT. Cells were resuspended in R buffer to $2.4 \times 10^7$ cells/ml. To deliver RNP into 16HBE14o- cells, The Neon Transfection System 10 µl Kit (Invitrogen) was used. Cas9 RNP complexes with or without ssODN donor template (2 µM) were electroporated into cells using the following conditions: 1,200 V, 20 ms, 4 pulses. After electroporation, the cells were transferred back to a medium without antibiotics. After 48 h, electroporation was validated by detecting ATTO550 signal of RNP with the Incucyte Live-cell analysis instrument (Sartorius).

## Analysis of Cas9/gRNA genome editing (cutting) efficiency in the pool of edited cells

Genomic DNA from 16HBE14o- cells electroporated with RNP complexes without ssODN template was extracted using Monarch Genomic DNA Purification Kit (New England Biolabs, NEB) according to the manufacturer's instructions. PCR was carried out with the following primers, forward: 5'-GCTCCTGCAGTTTCTAAAGAATATAG-3' and reverse: 5'-GAGAGGTATGACTGACCCATAAG-3' using Phusion High-Fidelity DNA Polymerase (NEB) to amplify a region of a 849-bp flanking target site of gRNAs in the fourth extracellular loop of WT-CFTR. PCR amplicons were purified with Monarch PCR and DNA Cleanup Kit (NEB). Purified amplicons were sequenced using Sanger sequencing with forward primer. To analyze the cutting efficiency of crRNAs, Sanger sequencing files from control (unedited) and edited cells were used as input into the TIDE web tool (Brinkman et al, 2014).

### ssODN templates for HiBiT knock-in position

Two variants of ssODN templates with 6 or 8AA linkers on either side of HiBiT (Table S1) specific for each crRNA were electroporated into cells along with Cas9/gRNA. After 7 d, pools of edited cells were seeded at three different concentrations: $2 \times 10^4$, $4 \times 10^4$, and $6 \times 10^4$ cells/well into CulturPlate-96, white opaque 96-well plate (PerkinElmer). The following day, lytic and extracellular assays were carried out, as mentioned below. The time-dependent decay of the luminescent signal for the lytic and extracellular assays was measured continuously for 3 h.

### Limiting dilution cloning

The pools of cells edited with Cas9/gRNA with ssODN (template for HiBiT tag knock-in) were used for limiting dilution cloning to obtain a monoclonal cell line. Cells were diluted in a conditioned medium to 1 cell/100 $\mu$l, transferred in individual wells of a 96-well plate, and grown for 3 wk. After ~7 d, plates were scanned for cell growth and to identify wells that contained only a single colony. Wells with more than a single colony were excluded from further experiments. After ~21 d (reaching ≥80% confluence), 3/4 of each well with a monoclonal cell line were transferred to 24-well plates, and 1/4 was used for primary screening.

### Primary and secondary screening for HiBiT-positive monoclonal cell lines identification

As mentioned above, in primary screening 1/4 of each well from the 96-well plate with the monoclonal cell line was transferred into CulturPlate-96, white opaque 96-well plate (PerkinElmer), and the lytic assay was carried out as mentioned below. Positive clones for the luminescent signal were retested after a week, using cells from 24-well plates, seeding $1 \times 10^4$ cells/well in triplicate for secondary screening, with the lytic and extracellular assays. Positive clones from both assays were expanded and cryopreserved.

### Luminescent assay to quantify the total level of HiBiT-tagged WT-CFTR protein (lytic assay)

To measure the expression of WT-CFTR-HiBiT protein in cells, Nano-Glo HiBiT Lytic Detection System (Promega) was used according to the manufacturer's protocol with a small alteration. Briefly, cells were seeded in 96-well plates or 384-well CulturePlate (PerkinElmer) and cultured for 24 h. The medium was removed before adding a mixed reagent containing lytic buffer with LgBiT, substrate, and fresh medium without FBS. The luminescent signal was measured by EnVision plate reader (Perkin Elmer).

### Luminescent assay to detect HiBiT-tagged WT-CFTR protein expressed on the cell surface (extracellular assay)

To measure membrane localization of WT-CFTR-HiBiT protein in cells, Nano-Glo HiBiT Extracellular Detection System (Promega) was used according to the manufacturer's protocol with a small alteration. Cells were seeded and incubated the same way as mentioned above in the lytic assay protocol. The medium was removed before adding a mixed reagent containing an assay buffer with LgBiT, substrate, and fresh medium without FBS. We optimized the total assay volume for both lytic and extracellular assays to 100 $\mu$l for 96-well plates and 25 $\mu$l for 384-well plates. The luminescent signal was measured by EnVision plate reader (Perkin Elmer).

### Heterozygotes and homozygotes genotyping, sequencing, and CRISPR/Cas9 off-target activity detection

Genomic DNA (gDNA) was extracted from HiBiT-positive clones from secondary screening using Monarch Genomic DNA Purification Kit (NEB) according to the manufacturer's instructions. PCR was carried out as mentioned in the section "Analysis of Cas9/gRNA genome editing (cutting) efficiency in the pool of edited cells." PCR amplicons were size-separated by 2% agarose gel electrophoresis. PCR amplicons from identified homozygotes were purified with Monarch PCR and DNA Cleanup Kit (NEB) and Sanger sequenced using forward primer: 5'- GCTCCTGCAGTTTCTAAAGAATATAG-3'. For detection of CRISPR/Cas9 off-target activity, gDNA was extracted and target sequences were amplified by PCR, as mentioned above, with primers listed in Table S2. Sanger sequencing was carried out only with forward primers.

### Western blotting and HiBiT blotting

For both assays, total protein lysates from $5 \times 10^6$ cells were prepared using ice-cold RIPA buffer (150 $\mu$M NaCl, 1% NP-40, 0.5% sodium deoxycholate, 0.1% SDS, 50 $\mu$M Tris–HCl pH 8.0 and 1 $\mu$M EDTA) supplemented with cOmplete Protease Inhibitor Cocktail (Roche). The lysates were incubated on ice for 30 min, vortexed, and centrifuged at 14,000$g$ for 30 min at 4°C. Protein concentration was determined by Pierce BCA Protein Assay Kit (Thermo Fisher Scientific), and 30 $\mu$g/well of total protein lysate were size-separated on 7.5% (wt/vol) SDS–PAGE gel. The resolved proteins were transferred to nitrocellulose membranes using the semi-dry blot method with the Trans-Blot Turbo Transfer System (Bio-Rad). **Western blotting**: the membranes were blocked with 5% nonfat dried milk in TBS buffer with 0.1% Tween-20 (TBST) for 1 h at RT, followed by incubation with the respective primary antibodies overnight at 4°C. The following day, membranes were washed in TBST and incubated for 1 h with peroxidase-conjugated secondary antibody (1:10,000; Sigma-Aldrich). A monoclonal anti-human CFTR antibody 596 (1:1,000; CF Foundation) was used to detect C and B bands of CFTR protein, and an anti-$\beta$-actin antibody (1:1,000; Sigma-Aldrich) was used as a loading control. Standard electrochemiluminescence-based detection was performed using ChemiDoc MP Imaging System (Bio-Rad Laboratories, Inc.) **HiBiT blotting**: Nano-Glo HiBiT Blotting System (Promega) was used for visualization of HiBiT-tagged proteins transferred on nitrocellulose membranes according to the manufacturer's protocol.

### Short-circuit current—Ussing chamber

Clone B38 and HBE cells were seeded at a $3 \times 10^5$ cells/cm$^2$ density onto 6.5 mm TranswellPermeable Supports (Corning Inc), and grown for 7 d in growth medium (100 $\mu$l Apical, 600 $\mu$l Basolateral), with medium changes every 2–3 d. Cells were grown in DMEM

(Wisent Inc) supplemented with 10% FBS (Wisent Inc), 1% non-essential amino acids (Sigma-Aldrich), 2 mM L-glutamine (Wisent Inc), 1 mM sodium pyruvate (Wisent Inc), 100 IU penicillin and 100 mg/ml streptomycin (Wisent Inc). Before electrophysiology analysis, cells were washed with PBS (Wisent Inc) and incubated in Opti-MEM Reduced Serum Medium (Thermo Fisher Scientific) supplemented with 10 mg/ml BSA (Sigma-Aldrich), (100 $\mu$l Apical, 500 $\mu$l Basolateral) for 72 h, with medium changes every 24 h. The short-circuit current ($I_{SC}$) was measured using an Ussing chamber (Physiologic Instruments). Trans-wells were mounted between two hemi-chambers containing 5 ml of basolateral or apical solutions, both at pH = 7.40. The apical solution consisted of 1.2 mM NaCl, 115 mM sodium gluconate, 25 mM NaHCO$_3$, 1.2 mM MgCl$_2$, 4 mM CaCl$_2$, 2.4 mM KH$_2$PO$_4$, 1.24 mM K$_2$HPO$_4$, and 10 mM D-dextrose. The basolateral solution consisted of 115 mM NaCl, 25 mM NaHCO$_3$, 1.2 mM MgCl$_2$, 1.2 mM CaCl$_2$, 2.4 mM KH$_2$PO$_4$, 1.24 mM K$_2$HPO$_4$, and 10 mM D-dextrose. Solutions in both hemi-chambers were maintained at 37°C and bubbled with 95% O$_2$ and 5% CO$_2$. Therein, monolayers were apically exposed to 10 $\mu$M amiloride (Sodium Channel blocker 1), 10 $\mu$M forskolin (cAMP agonist), 50 $\mu$M genistein (CFTR potentiator), 10 $\mu$M CFTR$_{Inh}$-172 (CFTR blocker), and 10 $\mu$M ATP (purinergic agonist to activate Ca$^{2+}$-activated Cl$^-$ channels) in this sequence (Gianotti et al, 2018). To assess barrier function, periodic 1 mV pulses were applied while recording the $I_{SC}$, and the resulting deflections in $I_{SC}$ pulse were used to calculate TEER. Monolayers with TEER ≥ 200 $\Omega \times cm^2$ were included in the analysis.

### Immunocytochemistry

Cells were seeded into a PhenoPlate 96-well plate (PerkinElmer) at 4 × 10$^4$ cells/well and incubated for 24 h at 37°C in a 5% CO$_2$. The cells were washed three times in PBS and incubated for 3 min with 2 $\mu$g/ml of WGA Alexa Fluor 555 (Invitrogen) in Hanks' Balanced Salt solution followed by 20 min fixation with 4% PFA at RT. In addition, cells were incubated for 15 min with 0.25% Triton X-100 in PBS for the condition with permeabilization. Subsequently, three washes with PBS were performed, followed by 90 min of blocking. A solution of 1% BSA in PBS was used for blocking and diluting primary and secondary antibodies. After blocking, primary anti-human CFTR monoclonal antibody TJA9 (1:250; CF Foundation) was added, and cells were incubated overnight at 4°C. The next day, all the wells were washed three times with PBS and incubated with anti-mouse Alexa Fluor 647 secondary antibody (1:1,000) for 60 min at RT and washed three more times. Finally, Hoechst 33342 in PBS (10 $\mu$M) was added for 10 min to the cells. Images were acquired using a Cell Voyager CV8000 high-throughput cellular imaging and discovery system (Yokogawa). Brightfield, Alexa Fluor 647 (ex 640; em 676/29 nm), WGA (ex 561; em 600/37 nm), and Hoechst 33342 (ex 405, em 445/45 nm) were captured by a 20x objective. All images were post-processed and analyzed using Columbus software version 2.7.1 (Perkin-Elmer) and ImageJ software.

### siRNA treatment and cell viability assay

siRNA transient transfection was performed using jetPRIME transfection reagent (Polyplus) following the manufacturer's protocol. For siRNA treatment, B38 cells were seeded in a 96-well plate (PerkinElmer) at 2 × 10$^4$ cells/well 1 d before transfection. CFTR Human siRNA Oligo Duplexes (Origene) variants in 10 and 50 nM concentrations were used for CFTR knockdown. Rab5 (RAB5A) and RAB11 Human siRNA Oligo Duplex (Origene) were used in 10 and 50 nM concentration for RAB5/11 knockdown. Transfection with a Universal Scrambled Negative Control siRNA (Origene) was used as a control. Cells were transfected for 48 or 72 h, followed by lytic and extracellular assay detection of CFTR levels. To exclude siRNA cell cytotoxicity an MTS assay was performed with a CellTiter 96 AQueous One Solution Cell Proliferation Assay (Promega) according to the manufacturer's protocol. The relative cell viability (%) was calculated as (A$_{sample}$ − A$_{background}$)/(A$_{control}$ − A$_{background}$) × 100%. The absorbance from the corresponding cell-free conditions was used as background.

### Brefeldin A treatment

B38 cells were seeded in the same manner as for siRNA treatment. Cells were treated for 6 h with Brefeldin A (Sigma-Aldrich) in concentration ranging from 1.56 to 25 $\mu$M followed by lytic, extracellular, and MTS assays.

### Statistical analysis

Statistical significance was calculated using one-way ANOVA—Tukey's test, ns = nonsignificant, *$P ≤ 0.05$, **$P < 0.005$, ***$P < 0.001$, using GraphPad Prism 8.0 software. In all tests, a $P$-value equal to or less than 0.05 was considered to be significant.

# Data Availability

The data that support the findings of this study are available from the corresponding author upon reasonable request.

# Supplementary Information

# Acknowledgements

We are grateful to the late Dieter Gruenert for providing the 16HBE14o- cell line, Martina Gentzsch (UNC-Chapel Hill, NC) and the CFFT Antibody Distribution Program for antibodies. This research was supported by The Quebec Ministry of Economy and Innovation (MEI; 625 MDEIE-PSVT3 to D Radzioch and JW Hanrahan), Ministry of School and Education of the Czech Republic (ENOCH CZ.02.1.01/0.0/0.0/16_019/0000868 and EATRIS-CZ LM2018133 to M Hajduch, D Radzioch, JB De Sanctis, and M Ondra); CIHR Grant FRN115117 #2433990; RI-MUHC Account 4925 to D Radzioch. All figures were created with BioRender.com.

### Author Contributions

M Ondra: data curation, formal analysis, investigation, methodology, and writing—original draft, review, and editing.

L Lenart: investigation, methodology, and writing—original draft, review, and editing.

A Centorame: investigation and writing—original draft, review, and editing.

DC Dumut: investigation and writing—original draft, review, and editing.

A He: investigation.

SSZ Zaidi: investigation.

JW Hanrahan: investigation.

JB De Sanctis: methodology and writing—original draft, review, and editing.

D Radzioch: conceptualization, supervision, investigation, and writing—original draft, review, and editing.

M Hajduch: conceptualization, data curation, supervision, funding acquisition, investigation, methodology, and writing original draft, review, and editing.

## Conflict of Interest Statement

The authors declare that they have no conflict of interest.

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
