## [Reviewer comments · Life Science Alliance]

Life Science Alliance

CRISPR/CAS9 BIOLUMINESCENCE-BASED ASSAY FOR MONITORING CFTR TRAFFICKING TO THE PLASMA MEMBRANE

Martin Ondra, Lukáš Lenart, Amanda Centorame, Daciana Dumut, Alexander He, John Hanrahan, Juan De Sanctis, Danuta Radzioch, Marian Hajduch, and Seyda Zaidi

DOI: <https://doi.org/10.26508/lsa.202302045>

Corresponding author(s): Marian Hajduch, Palacký University, Olomouc and Danuta Radzioch, McGill University

Review Timeline:

Submission Date:	2023-03-17
Editorial Decision:	2023-04-14
Revision Received:	2023-08-29
Editorial Decision:	2023-09-22
Revision Received:	2023-10-16
Editorial Decision:	2023-10-17
Revision Received:	2023-10-20
Accepted:	2023-10-20

Transaction Report:

April 14, 2023

Re: Life Science Alliance manuscript #LSA-2023-02045-T

Dr. Marian Hajdúch
Palacký University, Olomouc
Institute of Molecular and Translational Medicine
Hnevotinska 5
Olomouc 779 00
Czech Republic

Dear Dr. Hajdúch,

Thank you for submitting your manuscript entitled "CRISPR/CAS9 BIOLUMINESCENCE-BASED ASSAY FOR MONITORING CFTR TRAFFICKING TO THE PLASMA MEMBRANE" to Life Science Alliance. The manuscript was assessed by expert reviewers, whose comments are appended to this letter. We invite you to submit a revised manuscript addressing the Reviewer comments.

Thank you for this interesting contribution to Life Science Alliance. We are looking forward to receiving your revised manuscript.

Sincerely,

B. MANUSCRIPT ORGANIZATION AND FORMATTING:

Reviewer #1 (Comments to the Authors (Required)):

In this manuscript, Ondra et al develop a novel bioluminescence assay to monitor trafficking of CFTR based on the knock in of the HiBit peptide into the endogenous CFTR gene. The overall idea is interesting but there are some aspects that need to be clarified.

1. The use of a cell line that has endogenous expression is an obvious advantage. However, the authors fail to explain what the long term for developing such a tool is. It would be much more relevant if the used background would be mutant CFTR (e.g. F508del) with the aim of screening compounds to correct the trafficking defect. The developed tool (based on wt-CFTR) has a much more limited impact and the authors should elaborate a little bit more on the difficulties to create equivalent models expressing other CFTR variants - whether the strategy would be to insert the variants in the cell line produced here or to knock-in the HiBit into cell lines already expressing mutant CFTR in the same background (Valley et al 2019 and Santos et al 2023).
2. The quality of the figures is poor. The font size is rather small and when expanding to read what is shown, the figure appears quite blurred.
3. In Fig.3B and C, it looks that the best results are those obtained with Rnp3 (and not those obtained with the 8aa linker). Please clarify
4. The authors should check for off-target effects - predict them using one of the available online tools, and then test a couple of the most probable ones.
5. Page 9, 1st paragraph, line 4 - the authors are probably referring to 22 and not 32 B-clones.
6. Figure 5 WB results miss the adequate controls - unedited 16HBE cells should be tested in panel A. Furthermore, the authors should quantify the ratio of band C to total CFTR and assess if the knock-in affects the processing of CFTR. The full blot should be shown for inspection to confirm that in fact there are no "additional, prominent bands".
7. The immunofluorescence images in Fig.5B are of poor quality - and should include a membrane marker (co-localized with CFTR in e.g. non-permeabilized samples).
8. The Ussing Chamber analysis should be performed also using a non-edited control to ascertain that the knock-ins have equivalent activity of CFTR".
9. Fig.6 - the plots on "Total wtCFTR-HiBit" e "Membrane wtCFTR-HiBit" should use the same Y-axis to allow more accurate comparison.
10. To validate the ability of the assay to assess CFTR trafficking, controls should be tested - e.g. knock-down of genes that encode proteins essential for protein trafficking (select from the literature a couple whose effect has been shown on CFTR)
11. In the second and third paragraphs of the discussion, the authors explain how editing with CRISPR/Cas9 works and the role of RNPs and crRNA. This should appear earlier in the manuscript - probably when the strategy for the HiBit knock-in.

Reviewer #2 (Comments to the Authors (Required)):

Comments:

This manuscript deals with the development of a tool to using CRISPR/Cas9-based methodology to introduce a tag on the outer loop of CFTR (4th extracellular loop) and it allows them to determine total CFTR amounts and plasma membrane levels using a plate-based assay. The study lacks any novelty and does not advance the field.

Concerns:

1. The studies do not advance the field and is incremental. The study lacks novelty and several groups have already introduced tags to outer loop of CFTR. Tags include Flag, HA, HRP, fluorescent tags, etc., which allows various plate high-throughput screening assays that have been used for drug discovery. The authors have not demonstrated, how this method or reagent is superior or more innovative than several existing reagents?
2. There has been a lot of debate at the NACFC last year regarding the specificity of the CFTR antibodies. Specifically, the authors may want to validate some of the antibodies used in Fig 5 carefully.

Reviewer #3 (Comments to the Authors (Required)):

In the review by Ondra et al. they develop a novel tool to investigate surface levels of CFTR post-transit rescue. I have few comments.

- Figure 5B is confusing, why the permeabilization step should make the most of the CFTR staining intracellular for WT CFTR?
- 16HBEs are known to polarize poorly and show only weak function in Ussing chamber, based on the data shown in 5C, it is at three to four times less than the primary airways and CFBEos therefore, wondering whether this would be ideal as a test tool for function even there is a promising information based on the surface levels, it needs to be validated functionally.
- comparison with other tagging system e.g., simple antibody based is not presented to evaluate the efficiency of their tagged line.

August 28th, 2023

Attn: Dr. Eric Sawey, Life Science Alliance

RE: Manuscript entitled “**CRISPR/CAS9 BIOLUMINESCENCE-BASED ASSAY FOR MONITORING CFTR TRAFFICKING TO THE PLASMA MEMBRANE**” by **Ondra et al.** for consideration in **Life Science Alliance Journal**.

Ref: Submission of Revision #1

Dear Dr. Sawey,

We would like to thank you and the Reviewers for the thorough review of our manuscript.

We are submitting the first revision of the manuscript with the marked corrections [highlighted in yellow]. Important sections of the manuscript in which the changes were made are described below:

1. New version of Figure 5 with validated antibodies from Cystic Fibrosis Foundation.
2. New version of Figure 6 with all data normalized to control (% of control).
3. New Figure 7 with the interference of CFTR trafficking.
4. Figure S3 was deleted from the manuscript (data transferred to modified Figure 5).

We have answered the comments of the Reviewers as follows:

Reviewer #1: Comment 1:

In this manuscript, Ondra et al develop a novel bioluminescence assay to monitor trafficking of CFTR based on the knock in of the HiBit peptide into the endogenous CFTR gene. The overall idea is interesting but there are some aspects that need to be clarified. The use of a cell line that has endogenous expression is an obvious advantage. However, the authors fail to explain what the long term for developing such a tool is. It would be much more relevant if the used background would be mutant CFTR (e.g. F508del) with the aim of screening compounds to correct the trafficking defect. The developed tool (based on wt-CFTR) has a much more limited impact and the authors should elaborate a little bit more on the difficulties to create equivalent models expressing other CFTR variants - whether the strategy would be to insert the variants in the cell line produced here or to knock-in the HiBit into cell lines already expressing mutant CFTR in the same background (Valley et al 2019 and Santos et al 2023).

Answer to comment 1 of Reviewer#1:

We would like to thank Reviewer #1 for the thorough revision of our manuscript and the valuable insight offered. The main focus of this manuscript is the development of a molecular tool that can be used for further studies of CF pathology. Here, we first described the rationale and optimization process behind finding the best approach for the tagging of WT-CFTR with HiBiT tag. Moving forward, the scientific community can use our optimized tagging pipeline for tagging other CF variants (e.g., in the cell lines mentioned above) without any further optimization of crRNAs, ssODN templates, delivery mechanisms, or HiBiT position in the ECL4. Secondly, our developed cell line, B38, expressing WT-CFTR-HiBiT, can be used directly for assessment of the effect of pathogens or agents on WT-CFTR expression levels or membrane localization. Lastly, our ongoing efforts are indeed to develop cell lines with rare CFTR mutations tagged with HiBiT, allowing us to study the various molecular mechanisms underlying CF pathology.

Reviewer #1: Comment 2:

The quality of the figures is poor. The font size is rather small and when expanding to read what is shown, the figure appears quite blurred.

Answer to comment 2 of Reviewer#1:

The quality of the figures was only reduced for uploading the first submission. The font size was enlarged, full-resolution figures were attached and will be provided for publication purposes.

Reviewer #1: Comment 3:

In Fig.3B and C, it looks that the best results are those obtained with Rnp3 (and not those obtained with the 8aa linker). Please clarify.

Answer to comment 3 of Reviewer#1:

In Fig. 3B and C, we compared the effect of ssODN templates with 6AA or 8AA linkers around HiBiT in combination with RNP3 or RNP1 complexes for HiBiT knock-in by total or membrane detection of HiBiT tagged CFTR. When seeding cells at three different densities, templates with 8AA linkers provided a greater or similar signal compared with ssODN templates with 6AA linkers for HiBiT knock-in. The numbers highlighted in the graphs represent the signal for the highest seeding density of cells. These results demonstrate a higher luminescent signal for ssODN with 8AA linkers for RNP1 and RNP3. We did not use ssODN without any linkers for HiBiT knock-in.

Reviewer #1: Comment 4:

The authors should check for off-target effects - predict them using one of the available online tools, and then test a couple of the most probable ones.

Answer to comment 4 of Reviewer#1:

We used Chop-chop software for designing crRNAs suitable for CRISPR/Cas9 mediated HiBiT knock-in. Chop-Chop software predicted one possible off-target site for crRNA1 and four for crRNA3. All potential off-target sequences for crRNA3 (used for B38 clone preparation) have 3 mismatches in the corresponding crRNA sequence and are located in noncoding loci of the genome. Although there is a very low chance that these sites have been modified by CRISPR, based on the 3 mismatches, we plan to sequence the whole genome of B38 cells in the process of CRISPR preparation of other CF variants on the B38 cell line background.

Reviewer #1: Comment 5:

Page 9, 1st paragraph, line 4 - the authors are probably referring to 22 and not 32 B-clones.

Answer to comment 5 of Reviewer#1:

It was corrected in the manuscript.

Reviewer #1: Comment 6:

Figure 5 WB results miss the adequate controls - unedited 16HBE cells should be tested in panel A. Furthermore, the authors should quantify the ratio of band C to total CFTR and assess if the knock-in affects the processing of CFTR. The full blot should be shown for inspection to confirm that, in fact, there are no "additional, prominent bands".

Answer to comment 6 of Reviewer#1:

We added unedited control 16HBE14o cells (parental) to the experimental setup and replaced the WB immunoblot in Figure 5A. Furthermore, we enrolled in Cystic Fibrosis Foundation (CFF) CFTR Antibodies Distribution Program and used their fully validated antibodies: 596 for western blot analysis and TJA9 for immunocytochemistry. The full blot was attached during the resubmission as SourceDataF5.

Reviewer #1: Comment 7:

The immunofluorescence images in Fig.5B are of poor quality - and should include a membrane marker (co-localized with CFTR in e.g. non-permeabilized samples).

Answer to comment 7 of Reviewer #1:

The quality of images was only reduced for uploading the first submission. Full-resolution figures were attached for publication. We used wheat germ agglutinin (WGA) for staining the plasma membrane. As mentioned in comment 6, we used the CFF antibody TJA9 for IF.

Reviewer #1: Comment 8:

The Ussing Chamber analysis should be performed also using a non-edited control to ascertain that the knock-ins have equivalent activity of CFTR".

Answer to comment 8 of Reviewer #1:

The Ussing Chamber analysis of the non-edited 16HBE14o parental cells was performed and a representative Isc curve is shown in modified **Figure 5C**. While we observe higher Isc for 16HBE14o- cells compared with the B38 clone, there is still adequate response of the B38 clone to forskolin, genistein, and Inh172. Both 16HBE14o- and B38, provided a similar level of resistance during Ussing chamber analysis.

Reviewer #1: Comment 9:

Fig.6 - the plots on "Total wtCFTR-HiBiT" e "Membrane wtCFTR-HiBiT" should use the same Y-axis to allow more accurate comparison.

Answer to comment 9 of Reviewer #1:

To improve clarity, modified Fig. 6 was prepared with all siRNA results in plots normalized to control (non-treated cells).

Reviewer #1: Comment 10:

To validate the ability of the assay to assess CFTR trafficking, controls should be tested - e.g. knock-down of genes that encode proteins essential for protein trafficking (select from the literature a couple whose effect has been shown on CFTR).

Answer to comment 10 of Reviewer #1:

To confirm the ability of our assays to monitor CFTR trafficking, we used siRNA against RAB GTPases involved in endocytic CFTR trafficking. Rab5 facilitates the trafficking of CFTR from the cell surface to early endosomes, whereas Rab11 controls the trafficking of CFTR from recycling endosomes to the trans-Golgi network (TGN) and the trafficking of endocytosed CFTR back to the cell surface. In addition, we treated the cells with Brefeldin A to abrogate the transport of proteins from the endoplasmic reticulum to the Golgi apparatus. We inserted all the results into the new Figure 7 in the manuscript and discussed all the obtained results. In conclusion, our system was capable of monitoring changes in CFTR trafficking and membrane localization after its transport pathways were impaired.

Reviewer #1: Comment 11:

In the second and third paragraphs of the discussion, the authors explain how editing with CRISPR/Cas9 works and the role of RNPs and crRNA. This should appear earlier in the manuscript - probably when the strategy for the HiBiT knock-in.

Answer to comment 11 of Reviewer #1:

We would like to keep the second and third paragraphs of the discussion in the manuscript to keep the paper's coherence.

Reviewer #2: Comment 1:

This manuscript deals with the development of a tool to using CRISPR/Cas9-based methodology to introduce a tag on the outer loop of CFTR (4th extracellular loop) and it allows them to determine total CFTR amounts and plasma membrane levels using a plate-based assay. The study lacks any novelty and does not advance the field.

Concerns:

The studies do not advance the field and is incremental. The study lacks novelty and several groups have already introduced tags to outer loop of CFTR. Tags include Flag, HA, HRP, fluorescent tags, etc., which allows various plate high-throughput screening assays that have been used for drug discovery. The authors have not demonstrated, how this method or reagent is superior or more innovative than several existing reagents?

Answer to comment 1 of Reviewer#2:

We thank Reviewer#2 for the comments that prompted the revision of our manuscript. We have included points that highlight its novelty and superiority compared to the previously described CFTR protein tagging approaches. The novelty and advantages of our developed cell line with the HiBiT tag are the following:

- 1) This cell line expresses endogenous levels of WT-CFTR, rendering it a robust model, faithfully reproducing naturally occurring levels of CFTR. While most of the tagged CFTR systems were achieved through overexpression in various vectors, our system uses endogenously expressed CFTR preserving a natural state of the protein levels and balanced stoichiometry of protein complexes.
- 2) Our developed assay can easily be used in "Add and Read" HTS assays with fast single-reagent-addition and immediate detection of total or membrane-CFTR levels in live cells and in real-time.
- 3) Unlike some of the previous tools, this assay doesn't use multiple antibody binding steps and washes.
- 4) The B38 cell line expressing WT-CFTR-HiBiT was fully validated and can be used for further introduction of CFTR mutations. Unlike currently available cDNA based overexpression models, our model preserves natural gene organization and thus can be utilized for introduction of any type of CFTR mutations in future, including coding, non-coding and splice regions.

Overall, the most important advantages of our system compared to previously described methods of CFTR protein tagging are the simplicity of quantitative protein detection, high sensitivity, endogenous gene expression and its unmatched suitability for HTS screening for potential CFTR modifiers.

Reviewer #2: Comment 2:

There has been a lot of debate at the NACFC last year regarding the specificity of the CFTR antibodies. Specifically, the authors may want to validate some of the antibodies used in Fig 5 carefully.

Answer to comment 2 of Reviewer#2:

To address this issue, we enrolled in CFF CFTR Antibodies Distribution Program and used their validated antibodies 596 and TJA9, for western blot analysis and immunocytochemistry. We added non-modified 16HBE14o cells (parental) to the experimental setup and replaced the WB and IF figures in the manuscript.

Reviewer #3: Comment 1:

In the review by Ondra et al. they develop a novel tool to investigate surface levels of CFTR post-transit rescue. I have few comments.

Figure 5B is confusing, why the permeabilization step should make the most of the CFTR staining intracellular for WT CFTR?

Answer to comment 1 of Reviewer#3:

We express our gratitude to Reviewer #3 for the valuable feedback, which has led to the revision of our manuscript. Permeabilization allows the antibodies to penetrate the lipid membranes of the cell, and access the CFTR that is localized not only in the membrane but also in the cytoplasm. As a result, IF with permeabilization stains membrane and intracellular CFTR (total cellular CFTR), hence producing a higher fluorescent signal compared with IF without permeabilization.

Reviewer #3: Comment 2:

16HBEs are known to polarize poorly and show only weak function in Ussing chamber, based on the data shown in 5C, it is at three to four times less than the primary airways and CFBEos therefore, wondering whether this would be ideal as a test tool for function even there is a promising information based on the surface levels, it needs to be validated functionally.

Answer to comment 2 of Reviewer#3:

There are two widely used functional assays for CFTR: i) the Ussing chamber assay which measures ion movement across the cell membranes, and ii) the "patch clamp" assay, in which patches of the cell membrane are isolated using a micropipette tip and are hooked up to microelectrodes, which measure the opening and closing rates of single channels. The advantage of the Ussing chamber over the patch clamp studies is that it uses intact monolayers of cells and epithelia, not impaled with microelectrodes. As a result, the actual transepithelial current, as well as its direction, can be determined. It is a robust and sensitive technique for functional activity measurement where the influence of external stimuli is always compared to baseline current levels regardless of those being low or high. We measured and added the functional activity of 16HBE14o parental in Figure 5B, non-modified cells.

Reviewer #3: Comment 3:

Comparison with other tagging system e.g., simple antibody based is not presented to evaluate the efficiency of their tagged line.

Answer to comment 3 of Reviewer#3:

We compared expression of WT-CFTR in parental cell line (16HBE14o-) with WT-CFTR-HiBiT in clone B38 in Figure 5C using immunofluorescence staining.

We would like to thank the Editor and the Reviewers for their helpful and constructive suggestions which allowed us to greatly improve the quality of our manuscript. We believe that we have answered the Reviewer's comments satisfactorily and we hope that the Reviewers and the Editorial Board will find our revised manuscript acceptable for publication in *Life Science Alliance Journal*.

We are looking forward to your reply.

September 22, 2023

Re: Life Science Alliance manuscript #LSA-2023-02045-TR

Dr. Marian Hajduch
Palacký University, Olomouc
Institute of Molecular and Translational Medicine
Hnevotinska 5
Olomouc 772 00
Czech Republic

Dear Dr. Hajduch,

Thank you for submitting your revised manuscript entitled "CRISPR/CAS9 BIOLUMINESCENCE-BASED ASSAY FOR MONITORING CFTR TRAFFICKING TO THE PLASMA MEMBRANE" to Life Science Alliance. The manuscript has been seen by the original reviewers whose comments are appended below. While the reviewers continue to be overall positive about the work in terms of its suitability for Life Science Alliance, some important issues remain.

Our general policy is that papers are considered through only one revision cycle; however, given that the suggested changes are relatively minor, we are open to one additional short round of revision. Please note that I will expect to make a final decision without additional reviewer input upon re-submission.

Please submit the final revision within one month, along with a letter that includes a point by point response to the remaining reviewer comments.

To upload the revised version of your manuscript, please log in to your account: <https://lsa.msubmit.net/cgi-bin/main.plex>
You will be guided to complete the submission of your revised manuscript and to fill in all necessary information.

B. MANUSCRIPT ORGANIZATION AND FORMATTING:

Sincerely,

Reviewer #1 (Comments to the Authors (Required)):

Most of my concerns have been properly addressed. However, there are still a few that remain and should be answered before publication (numbering is kept from the previous round of revision).

4. Off target effects should in fact be checked at the current stage and not only as a "plan for the future" - as proof of principle, select one possible site and assess whether or not it is affected.

8. The authors should comment on the reduced CFTR activity of clone B38 compared to parental cells.

Reviewer #3 (Comments to the Authors (Required)):

Authors have made significant revision and very well addressed most of the concerns, however the quality of the figures is still sub-optimal, and they are hard to read because of the font size. Additionally, the confocal images in Fig. 5B are not satisfactory and staining needs some optimization. Do not have any other major concern.

October 7th, 2023

Attn: Dr. Eric Sawey, Life Science Alliance
RE: Manuscript entitled **CRISPR/CAS9 BIOLUMINESCENCE-BASED ASSAY FOR MONITORING CFTR TRAFFICKING TO THE PLASMA MEMBRANE** by **Ondra et al.** for consideration in **Life Science Alliance Journal**.

Ref: Submission of Revision #2

Dear Dr. Sawey,

We would like to thank you and the Reviewers for thoroughly reviewing our manuscript.

We are submitting the second revision of the manuscript with the marked corrections [highlighted in yellow]. We have answered the comments of the Reviewers as follows:

Reviewer #1:

Most of my concerns have been properly addressed. However, there are still a few that remain and should be answered before publication (numbering is kept from the previous round of revision).

Reviewer #1: Comment 4:

Off target effects should in fact be checked at the current stage and not only as a "plan for the future" - as proof of principle, select one possible site and assess whether or not it is affected.

Answer to comment 4 of Reviewer #1:

We checked all four potential off-target sites of crRNA3, used for insertion of the HiBiT tag in the B38 and A22 clone, by Sanger sequencing and did not detect any modification in these sites. This information was added to the manuscript together with sequences of primers used for the potential off-target site amplification and sequencing. As an example, we attached a chromatogram of one possible off-target site (genomic coordinates chr5: 3,196,548) below compared with the unmodified parental cell line (16HBE140-).

Reviewer #1: Comment 8:

The authors should comment on the reduced CFTR activity of clone B38 compared to parental cells.

Answer to comment 11 of Reviewer #1:

In Figure 5A, we observed that the B38 clone exhibits a lower expression of the CFTR protein compared to the parental cell line (16HBE14o-), likely due to a clonal selection process and heterogeneity of CFTR expression levels among cells in the 16HBE14o- cell line (Kerschner et al, 2023; <https://doi.org/10.14814/phy2.15700>). The reduced CFTR protein expression could account for the lower short-circuit current (ΔI_{Sc}) change observed in the B38 clone. However, it is noteworthy that despite this decrease, the B38 clone maintains an adequate response to forskolin (an activator) and Inh172 (an inhibitor) of CFTR. This suggests that the WT-CFTR-HiBiT in the B38 clone remains functional and responsive to these compounds, albeit in reduced quantities.

Reviewer #3: Comment 1:

Authors have made significant revision and very well addressed most of the concerns, however the quality of the figures is still sub-optimal, and they are hard to read because of the font size. Additionally, the confocal images in Fig. 5B are not satisfactory and staining needs some optimization. Do not have any other major concern.

Answer to comment 1 of Reviewer#3:

The font size was enlarged in Figures 3 and 4. Extra High Resolution figures 600 DPI were attached for publication as separate files the same way as it was for Revision #1. TJA9 anti-CFTR antibody used for immunofluorescence (IF) is a validated antibody from the Cystic Fibrosis Foundation Antibodies Distribution Program. The IF process was already thoroughly optimized. There is almost no background outside of the cells in confocal images. Furthermore, IF staining patterns for CFTR correspond to one in non-polarized bronchial epithelial cells as previously published.

We would like to thank the Editor and the Reviewers for their helpful and constructive suggestions which allowed us to improve significantly the quality of our manuscript. We believe that we have answered the Reviewer's comments satisfactorily and we hope that the Reviewers and the Editorial Board will find our revised manuscript acceptable for publication in *Life Science Alliance Journal*.

We are looking forward to your reply.

October 17, 2023

RE: Life Science Alliance Manuscript #LSA-2023-02045-TRR

Ms. Seyda Sadaf Zehra Zaidi
McGill University

Dear Dr. Zaidi,

Thank you for submitting your revised manuscript entitled "CRISPR/CAS9 BIOLUMINESCENCE-BASED ASSAY FOR MONITORING CFTR TRAFFICKING TO THE PLASMA MEMBRANE". We would be happy to publish your paper in Life Science Alliance pending final revisions necessary to meet our formatting guidelines.

- please add ORCID ID for corresponding and secondary corresponding authors--you should have received instructions on how to do so
- please remove the graphical abstract from the manuscript text and upload it separately with the file designation Graphical Abstract
- please exclude figures and their legends from the manuscript text
- please use the [10 author names et al.] format in your references (i.e., limit the author names to the first 10)
- please remove supplementary material from the manuscript text
- please upload all figure files as individual ones, including the supplementary figure files; all figure legends should only appear in the main text
- please add your main, supplementary figure, and table legends to the main manuscript text after the references section
- please add callouts for Figures S2A-B, S3A-D to your main manuscript text
- please add scale bars to Figure 5B

A. FINAL FILES:

B. MANUSCRIPT ORGANIZATION AND FORMATTING:

Sincerely,

October 20, 2023

RE: Life Science Alliance Manuscript #LSA-2023-02045-TRRR

Dr. Marian Hajduch
Palacký University, Olomouc
Institute of Molecular and Translational Medicine
Hnevotinska 5
Olomouc 772 00
Czech Republic

Dear Dr. Hajduch,

Thank you for submitting your Methods entitled "CRISPR/CAS9 BIOLUMINESCENCE-BASED ASSAY FOR MONITORING CFTR TRAFFICKING TO THE PLASMA MEMBRANE". It is a pleasure to let you know that your manuscript is now accepted for publication in Life Science Alliance. Congratulations on this interesting work.

DISTRIBUTION OF MATERIALS:

Again, congratulations on a very nice paper. I hope you found the review process to be constructive and are pleased with how the manuscript was handled editorially. We look forward to future exciting submissions from your lab.

Sincerely,
